# Direction Estimation in 3D Outdoor Air–Air Wireless Channels through Machine Learning

**DOI:** 10.3390/s23239524

**Published:** 2023-11-30

**Authors:** Muhammad Hashir Syed, Maninderpal Singh, Joseph Camp

**Affiliations:** Lyle School of Engineering, Southern Methodist University, Dallas, TX 75275, USA; mpvirdi@gmail.com (M.S.); camp@smu.edu (J.C.)

**Keywords:** directional communication, direction prediction, channel optimization, UAV Swarms, machine learning, 3D profiling

## Abstract

UAVs need to communicate along three dimensions (3D) with other aerial vehicles, ranging from above to below, and often need to connect to ground stations. However, wireless transmission in 3D space significantly dissipates power, often hindering the range required for these types of links. Directional transmission is one way to efficiently use available wireless channels to achieve the desired range. While multiple-input multiple-output (MIMO) systems can digitally steer the beam through channel matrix manipulation without needing directional awareness, the power resources required for operating multiple radios on a UAV are often logistically challenging. An alternative approach to streamline resources is the use of phased arrays to achieve directionality in the analog domain, but this requires beam sweeping and results in search-time delay. The complexity and search time can increase with the dynamic mobility pattern of the UAVs in aerial networks. However, if the direction of the receiver is known at the transmitter, the search time can be significantly reduced. In this work, multi-antenna channels between two UAVs in A2A links are analyzed, and based on these findings, an efficient machine learning-based method for estimating the direction of a transmitting node using channel estimates of 4 antennas (2 × 2 MIMO) is proposed. The performance of the proposed method is validated and verified through in-field drone-to-drone measurements. Findings indicate that the proposed method can estimate the direction of the transmitter in the A2A link with 86% accuracy. Further, the proposed direction estimation method is deployable for UAV-based massive MIMO systems to select the directional beam without the need to sweep or search for optimal communication performance.

## 1. Introduction

Unmanned aerial vehicles (UAVs) are finding their way into many day-to-day applications, but most of these applications currently require human intervention, either for drone flight or other roles. The scalability of any drone application lies in the capability to make autonomous decisions, either related to path planning or dynamic maneuvering in real-world deployments. Actions performed by drones depend on many factors, among which communication to and from drones is critically important. We now consider representative UAV applications and the challenges they bring.

**UAV Applications and Importance:** UAVs are primarily divided into two broader application types, i.e., military and non-military types [1]. In the initial years, airspace was tightly regulated, leading to the non-existence of drones in civilian domains. However, since 2006, when drones were allowed in civilian domains, there has been a significant increase in drone development, with newer applications emerging such as food deliveries, emergency first aid, and on-demand surveillance, among many others. Gartner [2] predicted that drone penetration in the delivery sector would reach a significant milestone of more than a million deliveries each day. In pursuit of making this prediction a reality, many researchers from academia and industry are contributing by solving various underlying challenges. One such challenge is accommodating a large number of drones in a small aerial space, considering the limited spectrum availability and the energy requirements, which necessitate optimizing the communication channels with the drones. Drones also have potential applications in future wireless networks, where they are envisioned to either support ground-based terrestrial networks or form an independent network serving ground users. In [3], a UAV-aided relay system is considered, where a UAV is deployed as a relay to support BS to communicate with a mobile device that is out of the coverage area. Likewise, in [4], the UAV is deployed as a relay in a device-to-device cellular system, where the objective of the UAV is to assist the communication link between nodes that suffer from direct communication links. In [5], we consider a drone that is deployed as an independent multiple-input multiple-output (MIMO) BS with full-duplex (FD) capabilities to support the communication needs of the FD ground users. Similarly, the authors in [6] studied multiple UAV-enabled multi-user communication models, where the altitude and antenna beamwidth of the UAV were optimized to improve the throughput. In addition, networks initially utilizing single UAVs are advancing by incorporating multiple UAVs. These UAVs distribute the computation and communication complexities among multiple aerial nodes while achieving a common task [7,8,9]. In [8], multiple UAVs are considered within a single-cell network, where the UAVs are deployed to offload collected user data to the BS either through a UAV-to-BS link when the SNR is high, or through the UAV–UAV link when the SNR is low. Subsequently, a joint optimization solution is proposed for the sub-channel assignment and trajectory planning of the UAV to maximize the uplink sum rate.

**UAV channels:** Drone communications have undergone several iterative improvements over the years. Initially, operators controlled drones through RF channels in a simplex mode, where commands to the drones were transmitted from the ground. Later, the communication channels evolved to duplex, allowing limited information flow in both directions. Due to low bandwidth, the feasible operational areas were restricted to a few hundred meters within the line of sight [10]. More recently, there has been a shift towards utilizing high bandwidth channels for applications requiring substantial data transfer, such as video sharing. For these purposes, channels like wireless local area networks (2.4 and 5 GHz) and 4G/5G networks are being employed today.

**Sense of Direction:** The use of directional antennas in wireless networks has been prevalent for a long time now. Some advantages of having a directional wireless network [11], which align well with the requirements of drone communications, include lower interference, improved spatial reuse, longer transmission range, and reduced power requirements [12]. In emerging wireless networks and massive MIMO systems, phased array antennas and beam steering techniques are utilized at the BS to focus RF energy toward ground users [13,14,15,16]. In such systems, the radiation space is usually divided into multiple small sub-spaces, and a narrow transmit beam, equal in size to the sub-spaced area, is steered across all the sub-spaces until the optimal beam, usually in the direction of the receiver node, is found. The major complexity typically arises from searching for the optimal beam, a task that is likely to become more challenging as the search space expands in 3D for communication between multiple UAVs. This complexity can be significantly reduced if the direction of the receiver is known at the transmitter. The direction-finding task can be performed through multiple methods, ranging from manually moving the directional antennas and listening to beacon signals, to using specialized profiling algorithms that assist in predicting the direction of the transmitter. On the basis of this, the direction-finding approaches have been studied in two subcategories as follows:*Conventional approaches*: The practice of finding the direction of wireless signals, such as RF, has been around for a long time and has found many applications in military and civilian domains. The basic idea is to move the directional receiver in all possible directions and observe the signal strengths. Once the point of the maximum signal strength is found, based on the SNR ratio, the receiver tries to estimate the direction of the transmitter to an approximate value. This approach is as simple as it sounds and, hence, has a very low cost of operation [12]. In the work by [17], the authors used an MAC protocol that transitioned from omnidirectional antennas to directional antennas, and they highlighted the advantages of the same. But, it obviously has an associated limitation, i.e., the time required to observe every possible point. For instance, if we consider an object that can move in a spherical motion area, then at the very least, it might have 360 points on each 2D plane. If the whole sphere is divided into N discrete 2D planes, then 360 × N possible planes arise. This results in the total number of possible combinations reaching millions, if not billions. Consequently, conventional non-smart methods fail to keep up with the speed and dynamic mobility requirements of modern drone applications such as vehicular swarms. Moreover, in multi-antenna systems, the direction of the node can be estimated using direction-of-arrival (DOA) techniques, such as MUSIC [18]. However, this requires snapshots of the signals at multiple antennas, leading to increased power consumption at the node. Additionally, in the presence of unknown signal sources, the estimation accuracy of traditional DOA methods is severely affected.*Intelligent approaches*: As discussed in the previous subsection, conventional approaches exist to estimate the direction of a wireless transmitter using characteristics of wireless signals, but most of these approaches are very time-consuming and, above all, not scalable to large systems. Further, as suggested in [19], machine learning (ML)-based intelligent approaches can be of great use in a wide range of applications, including gaining deeper insights into environmental features, such as channel dynamics, channel profiling, and user context awareness for quality of service. Moreover, with the data-driven capabilities of ML, optimizing wireless networks is possible. For instance, some studies, like in [20], have explored the use of directional antennas for wireless signal relaying, combined with reinforcement learning for predicting directions. In another work [21], the authors highlighted the importance of ML in modeling wireless channels. Similarly, in [22], the use of a deep learning-based approach is suggested for effectively predicting downlinks in massive MIMO scenarios. Additionally, in another work by [23], the authors suggest the use of ML in designing trajectories for the operability of multiple UAVs.

Further, wireless signals are greatly impacted by the surroundings in which they propagate. For instance, in closed indoor scenarios, the signals are expected to travel and traverse multiple paths due to the structures of walls and other indoor items. However, the signal propagation profiles tend to change drastically in open outdoor scenarios where there are fewer surfaces to deflect and reflect the signals. Many existing studies, such as those in [24,25], have focused on the indoor localization of wireless channels. However, outdoor environments remain less explored [26], especially in the context of focused transmissions on drones for efficient spectrum management and potential profiling. Motivated by these key research gaps, this research work aims to enhance the efficiency of UAV-to-everything communications in outdoor 3D spaces.

**Summary:** In this work, we study the communication link between two UAV nodes via an air–air (A2A) link and propose a novel learning-based direction-finding method that can be utilized at the receiver to estimate the direction of the transmit signal. For this purpose, we analyze the capacity and correlation performance of multi-antenna channels under A2A links in terms of azimuth and elevation angles between 2 MIMO UAV nodes. Then, we analyze the performance of the proposed method through empirical evaluations. The main contributions of the paper are summarized as follows:We propose a novel ML-based direction estimation method for the A2A link between two UAVs by designing the direction-finding problem as a classification problem utilizing the support vector machine (SVM) technique. The method utilizes multi-antenna correlation and capacity measures of the channel in an A2A link as features to estimate the direction of a UAV node. It achieves an accuracy of 86% when at least two RF chains with diverse antenna configurations are utilized at both the transmitter and receiver.We propose an application model of the ML-based direction estimation method for optimal beam selection in aerial massive MIMO systems and compare its complexity with the complexity of the conventional beam search algorithm. It is found that the proposed technique significantly reduces the computational complexity at the massive MIMO BS for the optimal directional beam selection when a large sum of antennas is utilized for transmission.

The rest of the paper is organized as follows. In Section 2, the measurement system and topology are discussed, which are utilized to characterize the A2A link and empirical evaluation of the direction estimation method. The equivalent communication model of the measurement system along with the performance metrics are discussed in Section 3. The proposed method for direction estimation and its application in the MIMO system is presented in Section 4. Section 5 presents the empirical characterization of the A2A link and the experimental evaluation of the proposed method. Finally, this paper is concluded in Section 6.

## 2. Experiment Setup

In this section, the multi-antenna software-defined radio (SDR)-based measurement system and measurement plan are presented. First, the implementation of the 2 × 2 Aerial channel sounding system is presented, which is utilized to capture MIMO A2A wireless channels. Second, the measurement topology design is presented, which is utilized to conduct outdoor wireless experiments.

### 2.1. UAVs Based 2 × 2 Wireless Channel Sounding System

We implemented a 2 × 2 wireless channel sounding system using two Universal Software Radio Peripheral (USRP) E312 units from Ettus Research [27], which were mounted on two commercial UAVs. The USRP E312 has two synchronized RF chains that are capable of receiving and transmitting in-phase and quadrature (IQ) signals. Note that USRP E312 can work in two distinct modes: *(i)* the embedded mode, where the USRP transmits and receives IQ samples standalone without external control from a host machine, and *(ii)* the network mode, where the USRP requires a host machine for transmission operations. Since our focus was on capturing UAV–UAV channels, we utilized the embedded mode of the USRP E312 due to its suitability for our experiments. Additionally, GNU Radio and USRP Hardware Driver libraries in Python were utilized to transmit and receive IQ samples.

We captured the wireless channel in one direction. Therefore, one of the UAV nodes worked as a transmitter and continuously transmitted two sinusoidal tones with frequencies of 5 and 15 KHz on RF chains TRXA and TRXB of the USRP E312 over a 2.484 GHz carrier frequency with a sampling rate of 200 kSamples/second and transmit power of +15 dBm, respectively. Both RF chains were equipped with an omnidirectional (VERT2450) antenna with 3 dBi of gain. However, the antenna on the TRXA chain was mounted horizontally while the antenna on the TRXB chain was mounted vertically. The transmitter USRP E312 was mounted on the top of the commercial UAV using custom-made 3D-printed parts to secure the SDR and antennas, as shown in Figure 1d. The other UAV node worked as a receiver, and we mounted the USRP and antennas on the receiver UAV node in an identical manner to the transmitter. The transmitted sinusoidal tones were continuously captured at two RF chains of the receiver USRP E312 during the flight of the UAV, and were post-processed in MATLAB offline. The received signal on each RF chain is sampled at 200 kSamples/second. Note that the UAV logs the flight details and the USRP captures the transmitted signal independently. Therefore, to synchronize the received IQ data with the UAV flight log, we recorded the start and stop times of the USRP script, and logged the data of the IMU sensor (i.e., roll, pitch, and yaw) of the USRP. The data synchronization method and timestamp calculation of the received IQ samples are detailed in [28].

### 2.2. Measurement Plan

We designed a measurement topology that comprises a total of 114 measurement points on the 3D sphere that is shown in Figure 1a, where the origin of the sphere is located at 80 m in altitude from the ground level and the radius of the sphere is 20 m. The transmitter drone hovers at the origin of the sphere while the receiver drone flies between the points on the sphere during the flight plan.

We utilized a spherical coordinate system, as shown in Figure 1b, to denote points on the 3D sphere. The 2D representation of the measurement topology in Figure 1a is presented in Figure 1c, where each measurement point is defined using azimuth and elevation angle pairs (Φ,θ). In Figure 1a, Φ is measured from the west axis and varies between 0 and 360 degrees in a clockwise manner, and θ is defined as the angle above or below the origin that varies between −90 (bottom of the sphere) and +90 (top of the sphere). We took measurements over 16 azimuth and 9 elevation angles at intervals of 22.5 degrees, totaling 114 locations on a 3D sphere. Note that at locations directly above or below the origin, i.e., θ = ±90 degrees, Φ is undefined. Moreover, the measurement points on the 3D sphere are translated into GPS coordinates using the mapping toolbox in MATLAB, exported to a keyhole markup language (.kml) file, and uploaded to the DJI GSPro automated flight planner tool. Note that the flight time of the UAV is limited by the capacity of the battery. Therefore, we designed a total of 8 flights (color-coded in Figure 1a) comprised of 16 measurement points each, where the receiver UAV node begins from the θ=0 degree (parallel to the origin), and flies in a vertical circle around the transmitter UAV node (origin) while hovering for 20 seconds at each elevation angle along the flight path. The headings of both the UAV nodes are set to the north during the flight time in each flight plan. Since both UAVs rely on the onboard GPS to maintain the position during the hover time at the measurement points on the sphere, we observed that the transmitter UAV node hovers at the origin of the sphere with an average standard deviation displacement error of 80 mm and, the receiver UAV node hovers at the measurement points on the sphere with a displacement error of 161 mm. These errors are negligible compared to the direct distance between the transmitter and receiver UAV nodes, which is 20 m.

## 3. Communication Model and Performance Metrics

In this section, first, the communication model of our channel sounding system is presented, which includes the estimation of the MIMO channel matrix. Second, wireless performance metrics are presented, which are utilized to characterize the MIMO A2A channel.

The equivalent communication model of our measurement system is depicted in Figure 2. Variables $x_{H}\left(k\right)=0.7e^{j2\pi f_{1}k}$ and $x_{V}\left(k\right)=0.7e^{j2\pi f_{2}k}$ are the two sinusoidal signal tones that are used for transmission from the two RF chains of the transmitter USRP, where $f_{1}=5KHz$, $f_{2}=15KHz$, and *k* is the discrete time index. Similarly, $y_{H}\left(k\right)$ and $y_{V}\left(k\right)$ represent the received signal at the two RF chains of the receiver USRP, which can be written as follows: (1)y(k)=Hx(k)+Γ(k),
where $y\left(k\right)={\left[y_{H}\left(k\right)\;y_{V}\left(k\right)\right]}^{T}$, $x\left(k\right)={\left[x_{H}\left(k\right)\;x_{V}\left(k\right)\right]}^{T}$, $\Gamma \left(k\right)={\left[\Gamma_{H}\left(k\right)\;\Gamma_{V}\left(n\right)\right]}^{T}$ represents the effective noise at the two receiver RF chains and H is the channel matrix, which can be defined as follows: (2)$$H=\left[\begin{array}{cc} h_{HH} & h_{HV}\\ h_{VH} & h_{VV} \end{array}\right],$$
where the norm of H satisfies $\left||H|\right|=\sqrt{2}$, $h_{HH}$ is the channel between the horizontal transmitter and horizontal receiver antenna, $h_{VV}$ is the channel between the vertical transmitter and vertical receiver antenna, $h_{HV}$ is the channel between the horizontal transmitter and vertical receiver antenna (diverse combination), and $h_{VH}$ is the channel between the vertical transmitter and horizontal receiver antenna (diverse combination). Next, we estimated the channel matrix H using the least square (LS) method as follows:(3)$$\hat{H}={\left(X^{H}X\right)}^{-1}X^{H}Y,$$
where $\hat{H}\in C^{2\times 2}$ is the LS estimate of the channel matrix, $X\in C^{2\times N}$ consists of *N* consecutive samples of the two transmit signal sinusoidal tones, $Y\in C^{2\times N}$ consists of *N* consecutive samples of the two received signal sinusoidal tones, and $X^{H}$ is the Hermitian transpose of X. We utilized N=100 samples for estimation as it provides a low mean squared-error performance. Also, for each measurement point on the 3D sphere, we calculated a total of 4000 consecutive channel matrix estimates using the IQ data captured by the receiver USRP. Note that the channel matrix estimation process is performed offline in MATLAB.

We are interested in analyzing the multi-antenna system performance of A2A channels. Therefore, we utilize two different types of performance metrics in three distinct multi-antenna communication system configurations to analyze the wireless performance over the 3D sphere. The multi-antenna system configurations in our measurement system are *(i)* 1 × 2 single-input multiple-output (SIMO), where one transmitter RF chain and two receiver RF chains are utilized; *(ii)* 2 × 1 multiple-input single-output (MISO), where two transmitter RF chains and one receiver RF chain are utilized; and *(iii)* 2 × 2 MIMO, where two transmitter RF chains and two receiver RF chains are utilized. The performance metrics are given as follows:

### 3.1. Capacity

We utilize the capacity of the channel as a performance metric to quantify the quality of the link in each multi-antenna system configuration. When channel state information (CSI) is available, the capacity per unit bandwidth for the SIMO system with the horizontal transmitter antenna can be calculated as [29]:(4)$$C_{SIMO-H}=E\left\{\log_{2}\left(\left|I_{n_{r}}\;+\frac{\zeta }{n_{t}}h_{SIMO-H}h_{SIMO-H}^{H}\right|\right)\right\}$$
where $\zeta =P/N_{o}$ denotes the average signal-to-noise ratio (SNR) at the receiver antenna elements, $n_{t}=1$ denotes the number of transmitter antennas, $n_{r}=2$ denotes the number of receiver antennas, I denotes the identity matrix, *E* denotes the expectation operator, ${\left\{.\right\}}^{H}$ denotes the conjugate transpose, and $h_{SIMO-H}={\left[h_{HH}\;h_{HV}\right]}^{T}$. For simplification, in this work, the SNR is normalized by setting P=1, which infers the measurement of the effective noise power in the absence of the transmit signal. Note that the capacity in the multi-antenna system depends on the knowledge of the channel at the nodes. Therefore, with the availability of channel information, multiple antennas in the system lead to independent parallel channels, and the channel capacity is the sum of the capacity of each spatial dimension. For large array antenna systems, the capacity improves linearly with the number of antennas. Similarly, the capacity per unit bandwidth for the SIMO system with the vertical transmitter antenna can be calculated as follows:(5)$$C_{SIMO-V}=E\left\{\log_{2}\left(\left|I_{n_{r}}\;+\frac{\zeta }{n_{t}}h_{SIMO-V}h_{SIMO-V}^{H}\right|\right)\right\},$$
where $h_{SIMO-V}={\left[h_{VH}\;h_{VV}\right]}^{T}$. The capacity per unit bandwidth for the MISO system with the horizontal receiver antenna can be calculated as follows:(6)$$C_{MISO-H}=E\left\{\log_{2}\left(\left|I_{n_{r}}\;+\frac{\zeta }{n_{t}}h_{MISO-H}h_{MISO-H}^{H}\right|\right)\right\},$$
where $n_{t}=2$, $n_{r}=1$ and $h_{MISO-H}={\left[h_{HH}\;h_{VH}\right]}^{T}$. Similarly, the capacity per unit bandwidth for the MISO system with the vertical receiver antenna can be written as follows:(7)$$C_{MISO-V}=E\left\{\log_{2}\left(\left|I_{n_{r}}\;+\frac{\zeta }{n_{t}}h_{MISO-V}h_{MISO-V}^{H}\right|\right)\right\},$$
where $h_{MISO-V}={\left[h_{HV}\;h_{VV}\right]}^{T}$. Last, the capacity per unit bandwidth of the MIMO system with $n_{t}=2$ and $n_{r}=2$ can be written as follows:(8)$$C_{MIMO}=E\left\{\log_{2}\left(\left|I_{n_{r}}\;+\frac{\zeta }{n_{t}}HH^{H}\right|\right)\right\}.$$

### 3.2. Correlation

We use the correlation between the channel estimate of the antennas on a UAV as a performance metric to quantify the system performance and efficiency of the antennas. For this purpose, we calculate intra-user correlation for different multi-antenna system configurations [30,31]. The correlation metric for the SIMO system can be described as the correlation between the receiver antenna elements and, mathematically, for the signals transmitted from the horizontal antenna of the transmitter, it can be written as follows:(9)$$R_{SIMO-H}=\frac{1}{n_{r}}\sqrt{E[|h_{SIMO-H}^{H}\times h_{SIMO-H}|]},$$
where $n_{r}=2$. Similarly, the correlation metric for the SIMO system utilizing the signals from the vertical antenna of the transmitter can be written as follows:(10)$$R_{SIMO-V}=\frac{1}{n_{r}}\sqrt{E[|h_{SIMO-V}^{H}\times h_{SIMO-V}|]}.$$

Subsequently, the correlation metric for the MISO system can be described as the correlation between the transmitter antenna elements. Therefore, the correlation between the signals captured at the horizontal antenna of the receiver can be written as follows:(11)$$R_{MISO-H}=\frac{1}{n_{t}}\sqrt{E[|h_{MISO-H}^{H}\times h_{MISO-H}|]},$$
where $n_{t}=2$. Similarly, the correlation metric for the MISO system with a vertical receiver antenna can be written as follows:(12)$$R_{MISO-V}=\frac{1}{n_{r}}\sqrt{E[|h_{MISO-V}^{H}\times h_{MISO-V}|]}.$$

Note that the value of *R* varies between 0 and 1 in all multi-antenna system configuration cases, where the R=0 infers that all the antenna elements are orthogonal with each other while R=1 shows that all the antenna elements are strongly correlated.

## 4. Proposed ML-Based Direction Estimation Method and Applications

In this section, we first present the proposed technique that can be utilized to estimate the direction of a node in the A2A link. Then, we discuss the application of the proposed technique in the massive MIMO communication nodes to achieve wireless performance gains with limited complexity.

### 4.1. Direction Estimation Using SVM

For direction estimation, the 3D sphere in Figure 1a is partitioned into eight different regions, where each region comprises several measurement points covering various azimuth and elevation angles, which are shown in Table 1 (note that the tops and bottoms of the regions are defined from the perspective of the location of the transmitter UAV at the origin of the sphere). The partitioning of the sphere in 2D is shown in Figure 3. The direction problem is approached as a supervised classification problem with a total of eight classes, where each region in the sphere denotes a quantized direction and represents a class. The size of the regions defines the complexity of the problem and the resolution of the localization. By increasing the number of classes and decreasing the size of each region, the complexity will be increased. Therefore, with eight classes in our classification problem, we propose that the width of the directional beam of the transmitter UAV should cover the entire region in the sphere, which gives leverage to the receiver UAV to optimize its position in the region based on other concurrent tasks, such as serving the users on the ground without sacrificing the A2A link quality.

In this work, the SVM model is adopted for the classification problem as it can provide better prediction accuracy with minimal training dataset, and it is deterministic in nature [32]. In general, the SVM is a binary classifier that is trained on a set of labeled training samples. For instance, let $\left(x_{i},y_{i}\right)\in R^{l}\times \left\{\pm 1\right\},i=1,\ldots ,N$ be a set of training samples with $x_{i}\in R^{l}$ being the input and $y_{i}\pm 1$ being the output. In order to train the SVM model, the data are classified through hyperplanes on the basis of their labels. The result of this process is a classifier decision function presented in Equation (Equation 13).
(13)$$f_{w,b}=\mathrm{sgn}\left(w\cdot x+b\right),$$
where *b* is the bias of the hyperplane, w is a coefficient vector, and sgn represents a bipolar sign function. The hyperplane of a classifier should meet the following conditions:(14)$$y_{i}\left[w\cdot x_{i}+b\right]\geq 1\;,\forall i=1,\ldots ,N.$$

Among all the separable hyperplanes that satisfy criterion (Equation 14), the separating hyperplane that has the greatest distance to the nearest point is referred to as the optimal separating hyperplane and will yield the optimal generalization. However, in many practical cases, the hyperplane may not be ideal. To allow for the possibility of violating criterion (Equation 14), it is possible to introduce some slack variables ($e_{i}\geq 0$ ) into (Equation 14), resulting in the following: (15)$$y_{i}\left[w\cdot x_{i}+b\right]\geq 1-e_{i}\;,\forall i=1,\ldots ,N.$$

Then, the training of SVM requires finding parameters w, x, and $e_{i}$, utilizing the following optimization problem:(16)$$\begin{array}{cc} \; & \min_{w,x,e_{i}}\;\frac{1}{2}w\cdot w+\frac{1}{2}C\sum_{i=1}^{N}e_{i}^{2}\\ \; & \;s.t.\;(15), \end{array}$$
where *C* is the constant parameter. Moreover, in order to solve non-linear problems, the data can be mapped to another dot product space F by a non-linear mapping $\phi :R^{N}\overset{}{\to }F$, after which, the above analysis can be performed in *F*. Therefore, for non-linear cases, problem (Equation 16) still holds with the constraint (Equation 15) being rewritten as $y_{i}\left[w\cdot \phi \left(x_{i}\right)+b\right]\geq 1-e_{i}\;,\forall i=1,\ldots ,N$, where ϕ is s function that maps the instances to higher dimensional spaces, and $K\left(x_{i},x_{j}\right)≜\phi \left(x_{i}\right)\cdot \phi \left(x_{j}\right)$ is the kernel function. In this work, we use a cubic polynomial kernel, which is defined as $K\left(x_{i},x_{j}\right)={\left(x_{i}\cdot x_{j}+1\right)}^{3}$.

We aim to use multiple features to predict the direction of the receiver node. In particular, the capacity and correlation metrics are considered as features of the A2A link in the SVM classification problem, which are defined in (Equation 4)–(Equation 12). Let *K* be the distinct available features extracted from the channel matrix H. It is, therefore, appropriate to have *K* separate SVM classifiers, which can be written as follows:(17)$$y_{j}\left(x\right)=f_{j}\left(x\right)=w_{j}\cdot \phi_{j}\left(x\right)+b_{j}\;,j=1,\ldots ,K.$$

Then, the output of the *K* features’ SVM classifier system for a given sample x can be written as: (18)$$f\left(x\right)=\sum_{j=1}^{K}f_{j}\left(x\right)=\sum_{j=1}^{K}w_{j}\cdot \phi_{j}\left(x\right)+b_{j}.$$

Note that all the base SVM classifiers are first trained to find the values of $w_{j}$ and $b_{j}$ using the labeled training samples, which are the capacity and correlation metrics in different regions of the sphere. Then, the label/quantized direction/sphere region of a testing sample can be directly determined by (Equation 18).

### 4.2. Application in Massive MIMO Systems

In general, massive MIMO systems are considered to be an important part of next-generation wireless networks as they can provide increased bandwidth and spectrum efficiency. In such systems, large antenna arrays are utilized with either fully digital or hybrid (analog and digital) beamforming architectures to achieve optimal communication performance. Subsequently, in the mm-wave band, the IEEE 802.11ad standard indicates high data rates (up to 7 Gbps) utilizing high gain antenna arrays with directional transmission techniques [13,14,15,16]. However, optimal directional gains can only be achieved when the receiver lies in the direction of the respective beams of the transmitter. In the IEEE 802.11ad standard, the search space for potential directional beams is partitioned, and the antenna radiation sphere is split into as many as 128 virtual sectors that can handle beam widths of less than 3 degrees. The best sector is selected by an exhaustive search, and the directionality gain is increased by fine-tuning the antennas at both transmission ends [16]. The complexity of searching for the best sector scales with the number of nodes and their sectors. For a single user, the exhaustive search complexity can be defined as $O(M^{2}+L^{2})$ [33], where *M* is the number of antennas and *L* is the number of auxiliary beams or sectors. In the case of aerial networks, the search space size will be greatly increased with UAVs flying at multiple altitudes, and the complexity is expected to increase compared to the ground networks. However, if the direction of the receiver is known at the transmitter, the optimal beam or sector can be easily selected, which will greatly reduce the search time and overall energy consumption.

For aerial networks with UAVs carrying massive MIMO nodes, the proposed ML-based direction estimation technique can be utilized to select the optimal directional beam for transmission between UAV nodes. The application model of the technique is given in Figure 4, where a UAV is shown to be carrying a massive MIMO node with a trained direction estimation SVM model and aims to select the optimal transmission beam toward the receiver UAV node (not shown) utilizing the features extracted from the CSI of the receiver. The accuracy of the model is periodically checked, and online training of the model is performed when the accuracy is less than a certain threshold. Note that the proposed model utilizes the channel estimates of only four antennas, reducing the communication overhead. The prediction complexity of the trained model can be defined as $O(n_{sv}f)$, where $n_{sv}$ is the number of support vectors and *f* is the number of features, which is significantly lower than the complexity of beam search techniques.

Subsequently, the benefit of using the proposed direction estimation technique over an exhaustive beam search can be shown by comparing the computation complexities. Since the proposed method predicts the quantized direction among eight quadrants of the 3D sphere, we compare the computational complexity of both methods with the total beam search space or sector L=8. The proposed method utilizes f=9 features and $n_{sv}=28$ support vectors, where the features are extracted from the channel estimates of only four antennas. The computational complexity of the exhaustive beam search method is calculated for an increasing number of BS antennas and is compared with the complexity of the proposed method in Figure 5, where the operations in the vertical axes correspond to the number of computation cycles required to estimate the correct beam sector or user direction. It is important to note that the complexity of the proposed method is independent of the number of antennas. The comparison shows the proposed method is computationally efficient for M>13, and for the BS with a large antenna array size (M=100), the proposed method has 38× less computational operations than the exhaustive beam search method. Similarly, the complexity is also calculated for different numbers of communicating users (*U*), which is represented in Figure 6, where M=150. The complexity in both methods increases with *U* due to the linear relationship of *U* with the number of operations, and the proposed method achieves significant performance improvement over the exhaustive beam search method. Also, for a large number of sectors in 3D space, the proposed method can be utilized in conjunction with the other beam search algorithms to reduce the beam search space size. For example, with the total number of beam sectors L=128, the proposed method can reduce the search space size to only 16 sectors with significantly lower computational operations than the exhaustive beam search method, resulting in reduced computation cycles on the BS. Therefore, for a large sum of antennas in the massive MIMO aerial node, the proposed method can provide a computationally efficient solution to select the direction of the transmission beams for communication performance improvements.

## 5. In-Field Experiment Results

We analyze the performance of the proposed SVM-based method for direction estimation in the A2A link over the 3D sphere by utilizing the measured correlation and capacity results in different multi-antenna configurations. Also, we aim to show the impact of adding RF chains in the system on the estimation accuracy of the proposed method. For this purpose, we analyze the performance of SVM in two cases: (i) the three-antenna system, where either one antenna is used at the receiver and two antennas are used at the transmitter, or vice versa, corresponding to either 1 × 2 SIMO configuration with the horizontal or vertical transmitter or the 2 × 1 MISO configuration with the horizontal or vertical receiver antenna. Note that we analyze all four combinations separately using the respective capacity and correlation results; (ii) four-antenna system, where both the transmitter and receiver utilize 2 antennas, and the system has the tendency to estimate capacity and correlation metrics in any possible multi-antenna configurations due to the availability of a 2 × 2 channel matrix. Moreover, we utilize a dataset consisting of 4000 realizations of the channel at each location on the 3D sphere. We perform 5-fold cross-validation on the dataset. The dataset is divided into two random subsets, where one of the sets is used as the testing data while the other is used as training data. We evaluate the performance of the proposed direction estimation method by measuring the prediction accuracy of the trained SVM model.

An example of the measured capacity and correlation results in the 1 × 2 SIMO system is shown in Figure 7, where the horizontal axis is the azimuth angle, the vertical axis is the elevation angle, the dark color shows the lowest correlation and capacity, and the light color shows the highest correlation and capacity. In the case of a horizontal transmitter antenna, the correlation values are on the lower side, mostly on the receiver locations between −45 and +45 degree elevation angles, while at the same locations, capacity values do not reduce significantly. Also, the capacity of the system is best when the receiver node is almost or exactly above and below the transmitter node i.e., from −67.5 to −90 and +67.5 to +90 degree elevation angles because these are the points where there is no UAV body obstruction between the channels. In the case of the vertical transmitter antenna, the results are almost opposite the horizontal antenna results. We found that the correlation values are significantly low overall when the receiver node is exactly above and below the transmitter node; this is because of the signal blockage due to the UAV body. Also, at the same locations, the capacity values are significantly low compared to the highest capacity results of the system achieved at other locations, which mostly lie between −45 and +45 degree elevation angles.

The performance of the proposed direction estimation method in the four-antenna system is shown through the confusion matrix in Table 2, where the total accuracy provides the percentage of correctly classified test data, and the class accuracy shows the percentage of correct labeling for each separate class of test data. The overall accuracy of predicting the correct region in the sphere is 86% and the individual class accuracy ranges between 66.2–95.9%. The inaccurate decisions mostly fall in the cross-diagonal and adjacent regions with respect to the true regions. For instance, in the case of SE-B, the incorrect decisions mostly lie in the NW-B region, which is the cross-diagonal region, and this is due to the identical characteristics of the channel in such regions that were not perfectly distinguished by the proposed method even with the two diverse antenna configurations at both the transmitter and receiver in the four-antenna system.

Next, we analyze the impact of adding an RF chain to the system by comparing the performance of the proposed method in the three-antenna and four-antenna systems. The total accuracy values of the proposed method in various multi-antenna configurations in the three-antenna system are calculated. It was found that the accuracy values range between 13.6 and 15.2%. Comparing the total accuracy values of the three- and four-antenna systems shows that the accuracy of the proposed method increased by at least 6.32× when adding an RF chain in the three-antenna system in both SIMO and MISO configurations and the low accuracy values infer that the proposed method cannot be utilized for direction estimation in the three-antenna system. The reason has to do with the limited ability of the features in the three-antenna system to distinguish between the different positions and regions in the 3D sphere. For instance, in the case of the 1 × 2 SIMO configuration, the capacity results with the vertical transmitter in Figure 7 do not significantly vary with respect to the azimuth and elevation angles at the receiver UAV locations that are parallel or almost parallel to the transmitter UAV (−22.5 to +22.5 degree elevation angles). Similarly, the capacity results with the horizontal transmitter in the same antenna configuration show limited variations when the receiver UAV flies at such locations that are normal or almost normal to the transmitter UAV (+67.5 to +90 and −67.5 to −90 degree elevation angles).

We also compare the performance of the proposed SVM method with other ML-based classification methods, such as decision trees, naive Bayes, artificial neural networks (ANNs), and k-nearest neighbor (KNN) [34]. For each method, we utilize the same features and dataset as in the proposed method for training and testing. In addition, we also evaluate the performance of the SVM method provided in [35] for direction estimation in the A2A link, where the authors in [35] utilized the received signal strength of the access point node as a feature in the SVM classifier for indoor localization. The direction estimation accuracy of each approach is presented in Table 3, along with the associated mathematical representation of prediction complexity and the number of operations required for predictions, where *T* is the depth of the tree in the decision tree method, $A_{n}$ and $A_{s}$ are the total number of layers and average number of neurons per layer in the ANN method, respectively, and *D* is the number of training samples in the KNN method. The comparisons show the superiority of the proposed SVM method in terms of accuracy. The SVM model from [35] provides the lowest accuracy of all because it only uses the received signal strength of a single communication path between the transmitter and receiver as a feature to estimate the direction. The KNN method provides comparable accuracy to the proposed method but at the cost of high complexity, which scales with the number of training samples. The decision tree method has the lowest complexity of all but provides insufficient accuracy.

Therefore, with the availability of a channel matrix at the nodes, the proposed method can be utilized to estimate the direction of the UAV in the A2A link with at least two diverse combinations of antennas at the transmitter and receiver. Moreover, the class accuracy and overall accuracy of the proposed method can be further improved by adding more RF chains and diverse combinations of antennas in the system, which can be easily imagined in massive MIMO UAV systems. The inclusion of additional antennas in the system can provide distinct communication performance over the sphere, aiding the proposed method in more efficiently distinguishing between classes, which is achieved by refining the support vectors based on new features. However, it is crucial to note that the complexity of the proposed method increases linearly with the number of features. Thus, it is desirable to incorporate such features in the proposed method that best contribute to identifying locations on the sphere, ultimately enhancing the accuracy of direction estimation.

## 6. Conclusions

In this work, we proposed a novel direction estimation method for the A2A link in a MIMO system that requires channel estimates of at least two diverse combinations of antennas at the transmitter and receiver. The proposed method uses the SVM technique which utilizes correlation and capacity measures of multi-antenna channels in the A2A link as features to estimate the quantized direction of the UAV node in a 3D sphere. For validation of the proposed technique, wireless parameters were measured between two UAV communication nodes, with each node carrying a USRP equipped with vertically and horizontally mounted antennas. The transmitter node hovers at the origin of a 3D sphere, while the receiver node flies at multiple points on the sphere, covering various azimuth and elevation angles. The results indicate that the proposed method estimates the direction of the UAV node with an accuracy rate of 86%, which supports the use of learning-based models with the available channel estimates to estimate the direction of the nodes in the A2A link. Moreover, this method for direction estimation can be directly utilized in UAV-based massive MIMO networks to efficiently select directional transmission beams without sweeping across multiple angles or partitioned sectors in 3D space, leading to reduced computational complexity and energy consumption at the aerial nodes.

## Figures and Tables

**Figure 1 sensors-23-09524-f001:**
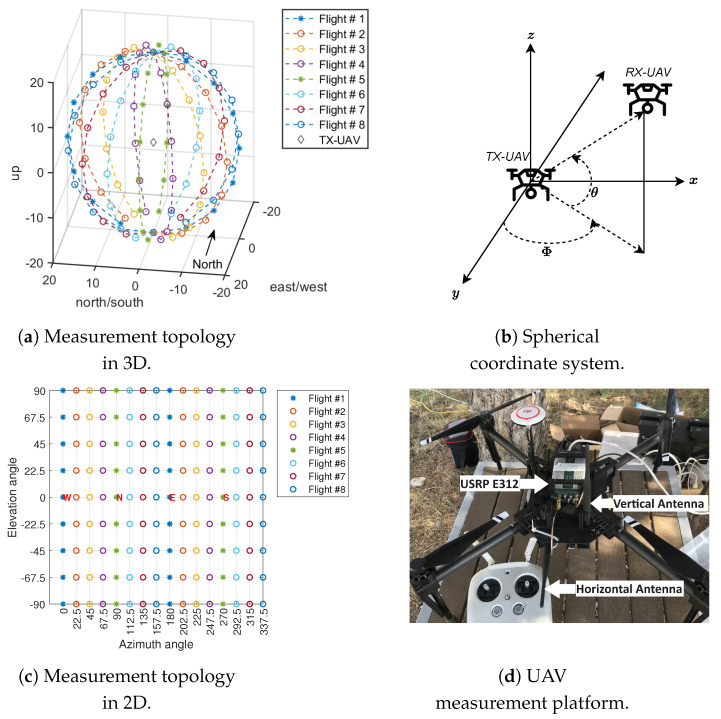
(**a**) The 3D sphere-based measurement topology, which comprises multiple flight plans covering various azimuth and elevation angle points, where the individual markers along the flight plan indicate receiver UAV positions. (**b**) The spherical coordinate system indicating azimuth and elevation angles between TX and RX UAVs. (**c**) The 2D representation of the sphere-based measurement topology with N, S, E, and W showing the north, south, east, and west points on the sphere, respectively. (**d**) UAV-based SDR communication setup with vertical and horizontal mounted antennas matched for TX/RX.

**Figure 2 sensors-23-09524-f002:**
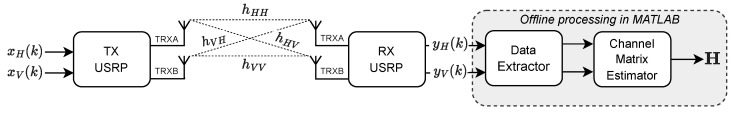
Communication model of the 2 × 2 MIMO channel measurement system.

**Figure 3 sensors-23-09524-f003:**
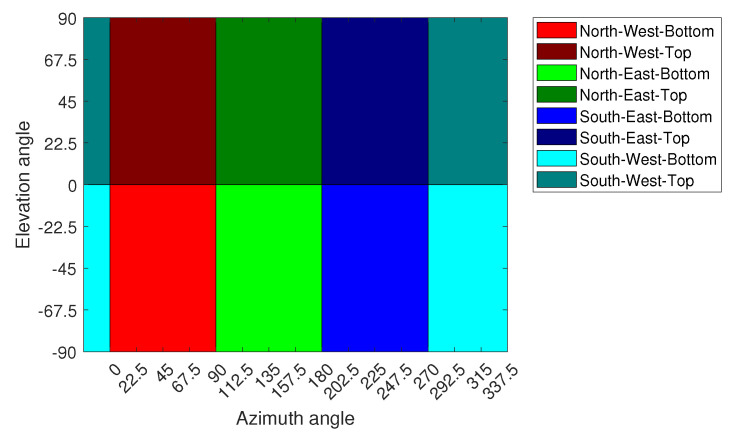
The 2D representation of partitioning of the sphere.

**Figure 4 sensors-23-09524-f004:**
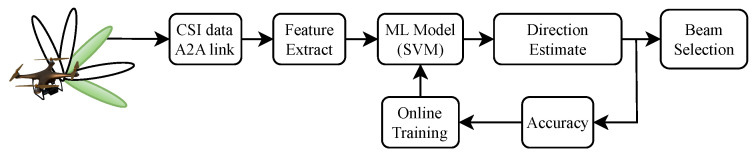
Application model of the proposed direction estimation technique for optimal beam selection.

**Figure 5 sensors-23-09524-f005:**
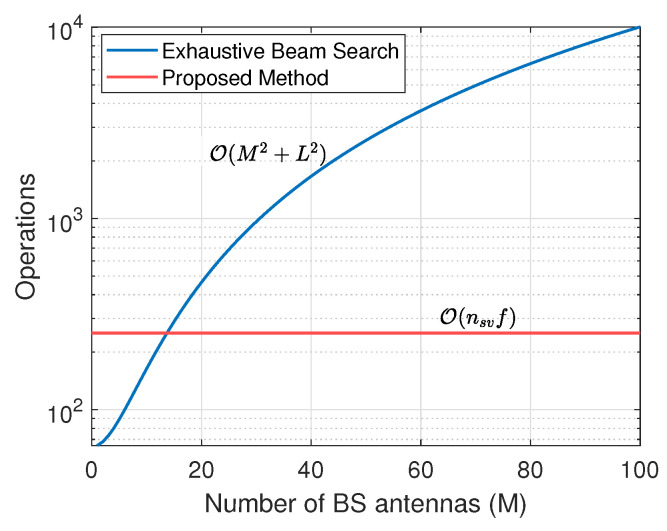
Exhaustive beam search complexity vs. the number of BS antennas along with the proposed method’s complexity.

**Figure 6 sensors-23-09524-f006:**
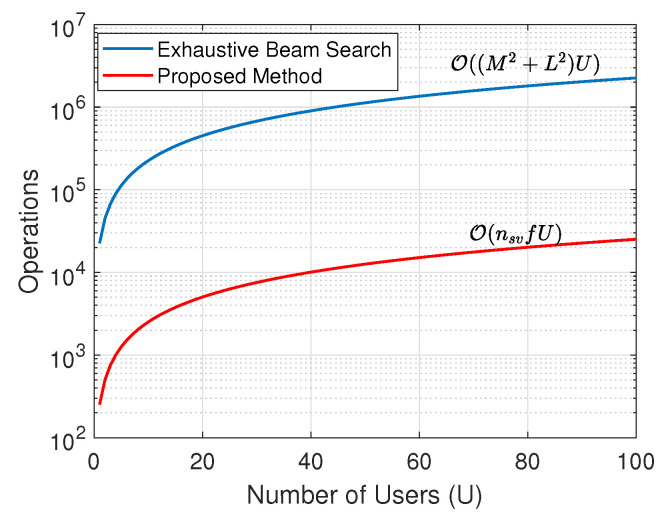
Exhaustive beam search and the proposed method’s complexities vs. number of users.

**Figure 7 sensors-23-09524-f007:**
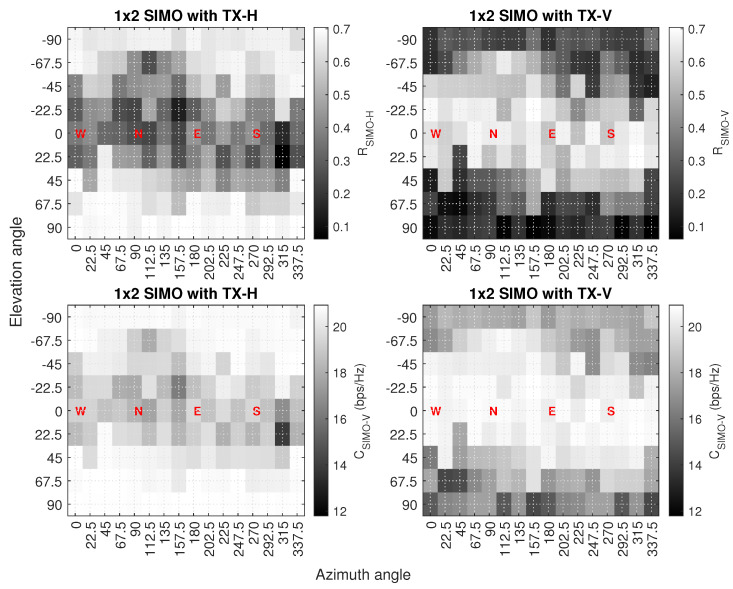
Capacity and correlation results in the 1 × 2 SIMO antenna configuration.

**Table 1 sensors-23-09524-t001:** Partitioning of 3D sphere into 8 regions.

Regions	Azimuth Angles	Elevation Angles
North-West-Top (NW-T)	0 to 90	0 to 90
North-West-Bottom (NW-B)	0 to 90	−90 to 0
North-East-Top (NE-T)	90 to 180	0 to 90
North-East-Bottom (NE-B)	90 to 180	−90 to 0
South-East-Top (SE-T)	180 to 270	0 to 90
South-East-Bottom (SE-B)	180 to 270	−90 to 0
South-West-Top (SW-T)	270 to 360	0 to 90
South-West-Bottom (SW-B)	270 to 360	−90 to 0

**Table 2 sensors-23-09524-t002:** Confusion matrix of the SVM in the 4-antenna system.

**Predicted** **Class**		**True Class**							
		**NE-B**	**NE-T**	**NW-B**	**NW-T**	**SE-B**	**SE-T**	**SW-B**	**SW-T**
	**NE-B**	1875	5	19	3	2		2	94
	**NE-T**	2	1403	51	2	4	1	63	74
	**NW-B**	120	72	1934	10	340	17	3	4
	**NW-T**	4	2	7	1885	9	79		14
	**SE-B**	1	14	310	120	1541	6	8	
	**SE-T**	1	4	9	40	9	1535	2	
	**SW-B**		62	8	4	3	3	1420	
	**SW-T**	58	328	7	4	4	2	2	795
**Class Accuracy**		93.8%	87.7%	77.4%	94.2%	77%	95.9%	94.7%	66.2%
**Total Accuracy**		86%							

**Table 3 sensors-23-09524-t003:** Performance comparison of the proposed method with various ML-based classification methods.

Method	Accuracy (%)	Complexity	No. of Operations
SVM [35]	10.2	$O(n_{sv})$	28
Naive Bayes	25.4	O(f)	8
Decision tree	53.0	O(log(T))	5
ANN	61.7	$O(A_{n}A_{s})$	10
KNN	78.7	O(Df)	4.6656 × 10${}^{5}$
Proposed method	86.0	$O(n_{sv}f)$	2.52 × 10${}^{2}$

## Data Availability

Data available in a publicly accessible repository that does not issue DOI. This data can be found here: https://smu.box.com/s/pnn69v22qv1hakz2xlcxw9puyhx906t9.

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
