# Peer review of "Direction Estimation in 3D Outdoor Air–Air Wireless Channels through Machine Learning"

_sensors, 2023, doi:10.3390/s23239524_

Round 1

Reviewer 1 Report

Comments and Suggestions for Authors

The paper deals with an interesting and up-to-date problem, and the way to find reliable solutions here will be important. Also, as such, the topic seems to grow in terms of gaining attention in the scientific and industry community. The paper is well structured and well written, it was quite easy to follow. The presented experiment looks fine and the results are fine with me. 

The way to improve the work are the following:

1. First, the title should be adjusted. The first impression is about spectrum utilization, but such a metric is not used in the paper (not widely enough); some definitions (math) of spectrum utilization should be provided and the way it is measured/calculated. Some results should show improvement in SU if the title remains unchanged. Maybe the removal of the first part would be fine (start with "Direction ...")

2. Section 3 is a bit superficial and it presents well-known relations; in fact, these relations should be adjusted accordingly depending on the applied MIMO processing scheme; As such, the section does not provide much insight and does not bring much value to the reader. 

3. On page 9, line 319; I would not agree that we have to split ALL points here (what about overfitting?); I mean in general it is true, yes, but this statement could be a bit too confusing, I would suggest rephrase or modification/explanation. 

4. Other than that, the work to me looks fine and complete. 

Reviewer 2 Report

Comments and Suggestions for Authors

The paper need to be improved

1.     How the proposed method will performance for higher order antennas.

2.     The method need to be compared with latest existing methods in terms of performance and complexity

3.     There are different wireless channels. Elaborate that what kind of channel model has been used.

4.     Why least square method has been used for channel estimation. How the method will perform for other estimation techniques like mmse

5.     For better understanding of the model, author should add more details about the approach like mathematical modeling, derivation in terms of capacity etc

6.     SVM technique was employed, author need to elaborate further about the performance in terms of various ML techniques.

Round 2

Reviewer 2 Report

Comments and Suggestions for Authors

Accept